# Crosstalk between BER and NHEJ in XRCC4-Deficient Cells Depending on hTERT Overexpression

**DOI:** 10.3390/ijms251910405

**Published:** 2024-09-27

**Authors:** Svetlana V. Sergeeva, Polina S. Loshchenova, Dmitry Yu. Oshchepkov, Konstantin E. Orishchenko

**Affiliations:** 1Institute of Cytology and Genetics, Russian Academy of Sciences, Lavrentieva 10, Novosibirsk 630090, Russia; polilos@bionet.nsc.ru (P.S.L.); keor@bionet.nsc.ru (K.E.O.); 2Department of Genetic Technologies, Novosibirsk State University, Pirogova 2, Novosibirsk 630090, Russia

**Keywords:** base excision repair (BER), non-homologous end joining (NHEJ), genome stability, NHEJ deficiency, transcription factor Sp1, transcription factor p53, p21 protein, cell cycle, scaffold protein XRCC4, hTERT overexpression

## Abstract

Targeting DNA repair pathways is an important strategy in anticancer therapy. However, the unrevealed interactions between different DNA repair systems may interfere with the desired therapeutic effect. Among DNA repair systems, BER and NHEJ protect genome integrity through the entire cell cycle. BER is involved in the repair of DNA base lesions and DNA single-strand breaks (SSBs), while NHEJ is responsible for the repair of DNA double-strand breaks (DSBs). Previously, we showed that BER deficiency leads to downregulation of NHEJ gene expression. Here, we studied BER’s response to NHEJ deficiency induced by knockdown of NHEJ scaffold protein XRCC4 and compared the knockdown effects in normal (TIG-1) and hTERT-modified cells (NBE1). We investigated the expression of the *XRCC1*, *LIG3,* and *APE1* genes of BER and *LIG4*; the *Ku70*/*Ku80* genes of NHEJ at the mRNA and protein levels; as well as *p53*, *Sp1* and *PARP1*. We found that, in both cell lines, XRCC4 knockdown leads to a decrease in the mRNA levels of both BER and NHEJ genes, though the effect on protein level is not uniform. XRCC4 knockdown caused an increase in p53 and Sp1 proteins, but caused G1/S delay only in normal cells. Despite the increased p53 protein, p21 did not significantly increase in NBE1 cells with overexpressed hTERT, and this correlated with the absence of G1/S delay in these cells. The data highlight the regulatory function of the XRCC4 scaffold protein and imply its connection to a transcriptional regulatory network or mRNA metabolism.

## 1. Introduction

Genome instability is a characteristic feature of cancer cells [1]. Although the question remains whether it is the cause or the consequence of malignant transformation, great efforts have been made to develop anticancer therapy relying on this fact [2,3,4]. DNA repair systems play a major role in maintaining genome stability; therefore, the proteins of these systems are used as targets for anticancer therapy [5,6,7,8,9]. Besides genome instability, 90% of cancer cells are also characterized by overexpression of the catalytic hTERT subunit of telomerase that is suppressed in normal cells [10,11,12,13,14,15,16,17,18,19]. Together, DNA repair pathways and the hTERT pathway are closely connected to cell cycle progression [20] and interact with each other in a complicated network of transcriptional regulation and protein post-translational modifications, as well as protein and mRNA metabolism. Many of these still-unrevealed interactions may interfere with the desired targeted effect.

The ultimate goal of any cancer therapy is to eliminate cancer cells, either to kill them directly or to cause programmed cell death, such as apoptosis or partanathos. Commonly used anticancer drugs aim to introduce DNA lesions that cause subsequent arrest of the replication system and kill highly proliferating cells. Targeting DNA repair pathways to suppress DNA repair, on one hand, can facilitate killing cancer cells via DNA-damaging agents, and on the other hand, it can promote synthetic lethality of cancer cells. Since DNA repair pathways are tightly connected to the cell cycle [21,22,23,24,25,26,27,28], by manipulating them, it may be possible to modulate cell cycle progression to drive the cell into programmed cell death. Thus, multiple effects can be achieved via targeting DNA repair pathways.

However, DNA repair pathways are an integral part of the entire cellular regulatory network, and affecting one triggers the response of different pathways. Furthermore, the overlapping redundancy of DNA repair may allow for compensation for the deficiency of originally targeted system by components of another DNA repair system [29]. Therefore, gaining insight into the crosstalk between repair pathways is particularly essential for anticancer therapies utilizing the components of DNA repair pathways either as the affected targets or as markers for prediction of treatment success. In certain cases, targeting DNA repair pathways could help to overcome cell resistance to DNA-damaging anticancer therapy [30,31,32]. Furthermore, almost all types of cancers already have some degree of DNA repair deficiency. Therefore, the knowledge of mutual interplay between different DNA repair pathways in the conditions of expression alterations is critical for the successful development of such therapies.

In human cells, there are five major DNA repair systems, i.e., base excision repair (BER), nucleotide excision repair (NER), mismatch repair (MMR), homologous recombination (HR), and non-homologous end joining (NHEJ), which protect the integrity of genomic DNA from the variety of possible endogenous and exogenous DNA-damaging factors. Two of the indicated repair pathways, BER and NHEJ, are highlighted, since they provide protection for DNA throughout the entire cell cycle [33]. BER is the major repair pathway that removes DNA single-strand breaks (SSBs), arising spontaneously due to the inherent instability of DNA [34,35]. Unrepaired SSBs promote cell cycle delay, which facilitates DNA repair prior to replication. NHEJ is the predominant form of double-strand break (DSB) repair in human somatic cells [36,37,38].

In general, each of DNA repair pathways includes a multi-protein complex assembled from a dedicated set of proteins for the particular DNA repair system that directly repairs the specific DNA lesion, and a number of factors involved in the regulation of DNA damage response. A particular DNA repair system does not require the components of the others to perform its function. However, changes in the amount of components of one DNA repair system can strongly affect the expression and efficacy of the others in vivo. Accumulating data indicate that different DNA repair systems interact with each other, and the question remains as to how they actually do this. On one hand, there is evidence that proteins from different DNA repair systems can directly interact with each other. On the other hand, being an integral part of the cell multifacet regulatory network, different DNA repair pathways share the same effectors, transcriptional factors, and signaling pathways, which can mediate the crosstalk between different DNA repair pathways [39]. For example, ubiquitous transcription factors Sp1 and p53, binding to similar DNA elements, participate in complex cellular regulation, including DNA damage response, stress response, and cell cycle progression [40,41,42,43,44,45,46].

Previously, we investigated how BER deficiency created by the knockdown of scaffold protein XRCC1 affects the expression of NHEJ system in immortalized NBE1 cells overexpressing hTERT (hTERT-modified NBE1 cells) [47]. We found that knockdown of the XRCC1 expression does not cause degradation of Sp1, but leads to downregulation of Lig4/XRCC4 and Ku70/80 at the mRNA and protein levels. We thus proposed the existence of an Sp1-independent backup mechanism that downregulates NHEJ in proliferating cells in response to BER deficiency.

Extending our previous investigation [47], we have studied the influence of NHEJ deficiency caused by knockdown of its scaffold protein XRCC4 on BER and NHEJ expression both at the RNA and protein levels, as well as on cell cycle progression, and compared their effects in normal TIG-1 [48] and hTERT-modified NBE1 cell lines [49,50].

## 2. Results

### 2.1. XRCC4 Knockdown Caused NHEJ Deficiency in Normal TIG-1 Cells

To create the experimental conditions of NHEJ deficiency in cells, we performed knockdown of scaffold protein XRCC4, which constitutes a platform for assembling proteins of the NHEJ complex into functional unit. We depleted XRCC4 by transfecting specific siRNA, which resulted in the inhibition of XRCC4 mRNA translation and verified the presence of knockdown after 72 h of transfection by evaluating mRNA and protein levels. Alignment of the XRCC4 siRNA sequence with mRNA sequences for the studied genes showed that they cannot be its targets. Blasting the used siRNA sequence against the RefSeq Select RNA sequences database for Homo sapiens (https://blast.ncbi.nlm.nih.gov/Blast.cgi (accessed on 14 September 2024)) produced the maximum lengths of the contiguous matching sequence for potential off-targets of only 13 bases, which demonstrated a negligible possibility of an off-target effect [51]. Knockdown of XRCC4 was very efficient in decreasing the relative levels of mRNA and protein down to 0.09 and 0.35, respectively, compared to the control group (Figure 1 and Appendix A).

To assess the degree of NHEJ impairment, we analysed the effect of XRCC4 knockdown on the expression of several participants of the NHEJ complex, namely, signaling proteins Ku70/ Ku80 and ligating enzyme Lig4. Lig4 functions exclusively in NHEJ, making it a central component of the repair process. Lig4 acts in complex with the XRCC4 that stimulates Lig4 enzyme activity in biochemical assays [52]. Together, XRCC4:LIG4 conduct the ligation of broken DNA strands, thus completing NHEJ.

We found that 72 h of XRCC4 knockdown differently affected protein and mRNA levels of Ku70/Ku80 heterodimer. The relative protein levels remained almost unchanged and were 1.15 for Ku70 and 1.07 for Ku80, while the mRNA levels decreased to 0.79 and 0.68 for Ku70 and Ku80, respectively. XRCC4 knockdown also led to a significant suppression of Lig4 expression both at the protein and mRNA levels (Figure 1). It is worth noting that knockdown affected the protein level more than mRNA, whose levels amounted to 0.18 and 0.48 relative units, respectively.

Thus, the observed results demonstrate that knockdown of scaffold protein XRCC4 induced alterations in the expression of the other NHEJ components. The knockdown effect manifested differently at the mRNA and protein levels. Noteworthy, knockdown caused a decrease in relative mRNA levels of all three analysed genes, Lig4, Ku70, and Ku80, while did not affect the levels of signalling proteins Ku70/Ku80, although it decreased the level of enzyme Lig4. Loss of either Lig4 or XRCC4 severely compromised NHEJ, resulting in NHEJ deficiency.

### 2.2. Effect of XRCC4 Knockdown on Expression of BER Components in Normal TIG-1 Cells

To determine how alterations in the NHEJ system impact the BER expression pattern, we analyzed the mRNA and protein levels of several key BER participants. We narrowed down the list of BER components to several representatives sufficient for evaluation of the BER functional status. From the list of BER components, we chose scaffold protein XRCC1 and enzymes Lig3 and APE1. All indicated proteins tightly cooperate in the multi-protein BER complex and are essential for the implementation of DNA repair via the BER pathway. Considering the crucial role of each of these proteins in the work of the BER complex, we believed that their expression levels would reflect the BER capacity.

Our experiments showed that knockdown of XRCC4 led to a decrease in the expression of all three analyzed genes both at the protein and mRNA levels (Figure 1). The relative decrease for scaffold protein XRCC1 and corresponding mRNA was not too pronounced and amounted to 0.77 and 0.88, respectively, compared to the untreated group. At the same time, XRCC4 knockdown more strongly affected the expression of enzymes Lig3 and APE1. The relative protein and mRNA levels of Lig3 were 0.54 and 0.61, respectively. The relative APE1 level decreased to 0.57 for protein and to 0.33 for mRNA.

The obtained data allow us to conclude that, in normal cells, knockdown of one NHEJ scaffold protein XRCC4 causes a subsequent decrease in the capacity of not only NHEJ, but of BER as well.

### 2.3. Effects of XRCC4 Knockdown on BER and NHEJ Expression in hTERT Modified NBE1 Cells

To elucidate if there are differences in BER and NHEJ response to NHEJ deficiency when hTERT is permanently overexpressed, we conducted a similar set of experiments with NBE1 cells (Figure 2 and Appendix A). Although being immortalized, NBE1 cells are considered normal cells, as they do not manifest phenotypic signs of cancer. However, these cells possess an elevated level of hTERT protein, and therefore, this might affect the expression pattern of genes related to the telomerase pathway.

First, we found that, in NBE1 cells, XRCC4 knockdown led to alterations in the expression of NHEJ components similar to the ones observed in normal TIG-1 cells. Specifically, the relative protein levels of Ku70/Ku80 did not significantly change, although the relative mRNA levels significantly decreased to values of 0.35 and 0.3, respectively. The relative protein level of Lig4 in NBE1 cells also decreased, as in TIG-1 cells (Figure 2), although to a smaller extent compared to TIG-1 (0.41 for NBE1 cells vs. 0.18 for TIG-1 cells). 

Regarding the BER system, we observed a slight but significant increase in the expression of the XRCC1 protein, although the mRNA level considerably decreased to 0.50 relative units. At the same time, Lig3 and APE1 protein expression levels decreased as in TIG-1 cells, but in NBE1 cells, the observed decrease was less pronounced. We also note that in NBE1 cells, XRCC4 knockdown had a greater suppressing effect on the mRNA level than in TIG-1 cells (Figure 2). This was observed for all analyzed BER and NHEJ genes.

Thus, XRCC4 knockdown caused NHEJ deficiency both in normal TIG-1 and hTERT-overexpressing NBE1 cells by inducing similar alterations in NHEJ components Lig4 and Ku70/Ku80. On the other hand, BER responded differently to XRCC4 knockdown in these two cell lines. The level of scaffold protein XRCC1, whose role is crucial for the BER pathway, manifested a tendency to slightly increase in hTERT-modified NBE1 cells, while decreasing in normal TIG-1 cells. It is important not to miss this fact, as this suggests the possible dependence of the XRCC1 protein level on hTERT.

### 2.4. Analysis of DNA Damage after XRCC4 Knockdown in Normal and hTERT-Modified Cells

We observed that knockdown of XRCC4 caused NHEJ and BER deficiencies. The existing knowledge suggests that such a state of DNA repair systems may cause DNA damage due to a permanent impact of endogenous damaging factors with a consequent DNA damage response. Therefore, we assessed DNA damage using the alkaline comet assay with subsequent analysis of cell images with Comet Analysis Software, Version 1.0.0.0 (Trevigen, Minneapolis, MN, USA). Out of the parameters that the software automatically calculates, we chose the DNA tail profile, as it is considered a more accurate parameter for assessing DNA damage [53]. We did not find signs of DNA damage either in normal or in hTERT-modified cells after 72 h of XRCC4 knockdown in our experimental conditions (Figure 3). This result is rather unexpected, as we found that XRCC4 knockdown disturbed the expression of key participants in the BER and NHEJ systems.

### 2.5. Effect of XRCC4 Knockdown on PARP1 Expression in Normal and hTERT-Modified Cells

For years, the main role of PARP1 was believed to be participation in DNA repair process and, specifically, in alternative NHEJ. Consequently, PARP1 was one of the primary targets of anticancer therapy aimed to inhibit DNA repair, and a host of inhibitors of PARP1 have been used in combination with DNA-damaging agents [54,55,56,57,58]. Indeed, PARP1 is activated by DNA single-strand breaks, and histone PARylation contributes to chromatin reorganization and recruitment of DNA repair complexes at DNA damage sites. It is also known that PARylation of XRCC1 prevents DNA damage-induced ubiquitylation [59]. Later, it became clear that PARP1-mediated PARylation serves to mediate a variety of pathways other than DNA repair [60,61,62,63,64].

We analysed *PARP1* expression after XRCC4 knockdown and found that PARP1 protein levels increased in both cell lines to 2.03 and 2.37 relative levels for TIG-1 and NBE1 cells, respectively (Figure 4, Appendix A).

### 2.6. The Effect of XRCC4 Knockdown on Transcription Factors p53 and Sp1 in Normal and hTERT Modified Cells

Since knockdown of one scaffold protein of the NHEJ system led to subsequent simultaneous multiple alterations in the expression of both repair systems, BER and NHEJ, it apparently triggered a multifaceted, finely regulated cellular response. Therefore, we analysed the expression of two ubiquitous transcriptional factors Sp1 and p53, which are highly responsive to stress stimuli and play surveillance roles in the coordination of all these pathways, including DNA damage response and cell cycle decisions.

We found that p53 responded to XRCC4 knockdown with an elevated protein level in both cell lines (to 1.48 and 1.83 relative levels for TIG-1 and NBE1 cells, respectively) (Figure 5 and Appendix A), but this elevation did not result from transcription up-regulation, since in both cell lines, XRCC4 knockdown caused a significant decrease in mRNA levels. Most probably, the observed elevation of p53 protein originated from either its stabilization or proteolysis inhibition. Altogether, the oppositely directed changes observed in p53 mRNA and protein levels are consistent with the existing knowledge about maintaining the cellular amount of p53 protein under normal and stress conditions, and most probably indicate the alterations in the p53 proteolysis pathway.

XRCC4 knockdown caused a similarly significant elevation of Sp1 protein levels in both cell lines to relative values of 3.15 in TIG-1 and 3.12 in NBE1 cells. At the same time, knockdown differently affected Sp1 mRNA levels depending on the cell line. More specifically, in normal TIG-1 cells, knockdown caused a slight but significant mRNA level increase to 1.39 relative units, while in hTERT-modified cells, knockdown led to considerable suppression of the mRNA level to 0.55 relative units.

### 2.7. Cell Cycle Alterations after XRCC4 Knockdown in Normal and hTERT-Modified Cells

It is known that, in normal cells, DNA damage causes G1/S delay or cell cycle arrest and apoptosis depending on the severity of DNA damage, and this is regulated by the p53 protein. However, the level of p53 protein does not ultimately depend on DNA damage. Creating the conditions of NHEJ deficiency, we expected that this would cause a DNA damage response and G1 phase delay. Therefore, we evaluated the impact of XRCC4 knockdown on cell cycle progression by estimating the percentage distribution of cells throughout the cell cycle phases using FACS analysis with propidium iodide staining.

The analysis of normal TIG-1 cells showed that the main part of the whole cell population of control and XRCC4 knockdown groups accumulated in the G1 phase of the cell cycle, although the percentage of G1 population in knockdown group was slightly higher than in the control group and was 81% and 74%, respectively. Consistent with this observation, the number of cells in the S phase decreased almost twofold after XRCC4 knockdown compared to control cells. These changes in the G1 and S phases demonstrate that XRCC4 knockdown slightly slowed down the G1 phase of the cell cycle and delayed cells entering into the next S phase. Further, we did not observe significant differences in the G2/M subpopulation between the control and knockdown groups, the percentages being 15% and 14%, respectively.

The comparison of cell cycle progression after XRCC4 knockdown in normal and hTERT-modified NBE1 cells revealed several differences. In hTERT-modified cells, like in normal TIG-1 cells, most of the whole cell population accumulated in the G1 phase. However, in NBE1 cells, XRCC4 knockdown led to a small but significant decrease in the G1 population (84% for control vs. 72% for XRCC4 knockdown groups) and a slight but significant increase in cell percentage in the S phase (5% for control vs. 11% for XRCC4 knockdown groups), and these changes are opposite to the ones observed in the TIG-1 cells. Thus, the XRCC4 decrease promoted G1/S transition in hTERT-modified cells. Additionally, we noted that, in hTERT-modified cells, the number of G1 cells was slightly but significantly higher compared to normal: 84% vs. 75% for NBE1 and TIG-1 cells, respectively. We also noted a considerable difference between percentage values of the G2/M cell subpopulation between control groups of normal and hTERT-modified cells. The size of the G2/M population in NBE1 cells (6%) was nearly three times smaller than in TIG1 cells (15%). XRCC4 knockdown caused a slight increase in the G2/M subpopulation from 6% in the control to 10% in the XRCC4 knockdown groups in NBE1 cells. Also, a small sG1 subpopulation was registered for NBE1 cells, and the proportion of this population slightly increased after XRCC4 knockdown from 5% to 7% (Figure 6).

We also assessed the expression of the p21 protein after XRCC4 knockdown in normal and hTERT-modified cells. This protein plays an important role in cell cycle transition from the G1 to the S phase and can cause cell cycle arrest if it binds to the CDK2/CYCE complex. Transcription of p21 is positively regulated by the p53 transcription factor. We found a slight difference in the p21 response to XRCC4 knockdown between the cell lines. In normal TIG-1 cells, p21 significantly increased to a 1.69 relative protein level, while in hTERT-modified NBE1 cells, there was only a tendency to increase, and it was not significant and amounted to a 1.17 relative protein level (Figure 7, Appendix A). Consequently, the effect of XRCC4 on G1 phase delay should be less pronounced in hTERT-modified cells compared to normal cells. This is indeed what we observed in the cell cycle analysis described above.

## 3. Discussion

In the present study, we wished to gain more insight into the crosstalk between BER and NHEJ repair pathways and their mutual effect on cell cycle progression in the conditions of NHEJ deficiency. With this in mind, we modelled NHEJ deficiency by knocking down its scaffold protein, XRCC4 and compared the resulting effects in normal TIG-1 cells and in immortalized hTERT-modified cells NBE1. A summary of our experimental results is given in Figure 1. Although NBE1 cells are considered normal cells, they have overexpressed hTERT subunits, which may provide important data on the BER response to NHEJ deficiency against the background of overexpressed hTERT simulating crosstalk between BER and NHEJ repair systems in cancer cells. Another feature of NBE1 cells is genome instability, which is inherent for them before immortalization, including trisomy in chromosome 19, where gene XRCC1 is located.

XRCC4 knockdown caused NHEJ deficiency via a significant decrease in its key components, i.e., XRCC4 itself and Lig4, on protein and mRNA levels in both cell lines. The loss of either Lig4 or XRCC4 severely compromises NHEJ. There is evidence that Lig4 is unable to function without its constitutive interaction partner XRCC4 in cells and becomes unstable when it is missing [65,66,67,68,69,70]. It was also previously reported that the level of Lig4 protein can be altered not necessarily due to a change in the gene expression, but rather due to protein degradation in the absence of XRCC4 [70]. This may have played a role in our experiments, and the observed decrease in the Lig4 protein after knockdown more probably originated from protein turnover rather than decreased mRNA levels. This is in line with the generally known higher rates of protein turnover, which can provide a more rapid response to stress stimuli than the corresponding transcription pathway. It is worth noting that NHEJ deficiency at the protein level does not necessarily cause crucial DNA damage in the absence of strong DNA-damaging factors like H_2_O_2_ or ionization radiation. In fact, in our study, we did not find significant differences in total DNA damage between the knockdown and control groups in the alkaline comet assay either inTIG-1 or in NBE1 cells.

Characterizing the effect of XRCC4 knockdown on the other components of NHEJ, we noted that the levels of signalling proteins Ku70/80 did not change, while their mRNA levels significantly decreased, and the effect was similar in both normal and hTERT-modified cells. This is rather remarkable, and the possible explanation is the known high abundance of Ku70/80 heterodimer in cells, which allows it to quickly respond to a break and promote the binding of XRCC4/Lig4 to DNA ends [71,72]. Comparing these results with previously obtained data on the knockdown effect of the BER scaffold protein XRCC1 on the NHEJ system, we further note that a decrease in the XRCC1 protein resulted in a decrease in the Ku70/80 heterodimer in NBE1 and TIG-1 cells. Thus, the level of Ku70/Ku80 protein is more sensitive to the XRCC1 level than to the XRCC4 level. Furthermore, it is known that the promoter regions of *Ku70/Ku80* genes have Sp1 binding sites, and the expression has been reported to be regulated by Sp1. Our data showed that, despite an increase in the level of Sp1 protein, Ku70/Ku80 expression decreased at both the protein and mRNA levels, which implies the involvement of other transcription factors.

Turning to crosstalk between BER and NHEJ systems, XRCC4 knockdown also caused the conditions of BER deficiency in both cell lines. This was revealed by a significant decrease in mRNA and protein levels of the key BER enzymes, i.e., APE1 and Lig3. It is important to note that the BER scaffold protein XRCC1, a crucial participant in the multiprotein complex, responded to the decrease in the NHEJ scaffold protein XRCC4 differently depending on the cell line. In normal TIG-1 cells, XRCC4 knockdown decreased mRNA and protein levels of XRCC1, while in hTERT-modified NBE1 cells, the protein level slightly increased against the background of a considerably decreased, almost halved mRNA level. This suggests that hTERT may affect XRCC1 protein metabolism. Notably, in our previous study, we found that XRCC1 knockdown caused a decrease in XRCC4 protein levels in both normal TIG-1 and hTERT-modified NBE1 cells. Thus, there is indeed bidirectional crosstalk between the two repair systems.

Taken together, these data show that in normal cells, when the expression of hTERT is suppressed and, consequently, the hTERT protein level is very low, the protein levels of both scaffolds depend on each other and a decrease in one is followed by a decrease in another. Most probably, this is regulated indirectly via intermediate pathways along the XRCC4-XRCC1 axis. On the other hand, in hTERT-modified cells, when hTERT is overexpressed, the pathway starting from XRCC4 decrease and leading to XRCC1 decrease is interrupted by the hTETR protein, indicating the dependence of XRCC1 on hTERT, rather than on XRCC4 protein decrease. Perhaps, in normal cells, this dependence does not manifest itself because the level of hTERT is low.

Thus, we see that knockdown of one NHEJ scaffold protein, XRCC4, caused deficiency in both repair systems, BER and NHEJ, and in both normal and hTERT-modified cells. Remarkably, a decrease in XRCC4 protein decreased mRNA levels for all analysed genes of both repair systems, while its effect on the expression of the corresponding proteins manifested differently. Regarding proteins, this can be rationalized by the existence of differing pathways regulating their quantitative levels, e.g., by targeting proteins to proteasomal degradation via post-translational modifications in response to changes in extracellular and intracellular conditions.

The question is why the decrease in the XRCC4 protein caused mRNA decreases in the other genes of BER and NHEJ systems? The obtained data are not sufficient to reconstruct the sequence of events starting from the decreasing protein XRCC4 in cytosol and leading to a decrease in mRNA level because this response involves multiple steps and numerous mediators. Cellular mRNA levels are determined by the rates of mRNA synthesis and degradation. We believe that we cannot definitely tell whether the decrease originated from transcriptional suppression or overlapping mRNA degradation via the intrinsic mRNA metabolism [73,74,75].

However, the available data allow us to suggest a possible connection between XRCC4 and the discussed decrease in mRNA levels. XRCC4 belongs to the class of scaffold proteins [76,77,78,79]. While the term scaffold implies a static, supportive platform to organize multiple proteins into efficient functional units, like in the case of multi- BER and NHEJ protein complexes, some of the scaffolds are known to be key regulators of many signaling cascades [80,81,82]. Furthermore, scaffold proteins can form signalosomes, large supramolecular protein complexes which enhance signaling efficiency by increasing local concentration and signaling activity of the individual components, and this type of regulation has been reported for DNA repair.

It is generally known that XRCC4 protein functions in NHEJ as a hub to arrange an efficient multiprotein functional unit to repair double-strand breaks [83]. However, other functions of XRCC4 are also considered, although they have not yet been fully understood. We believe that it may participate in alternatives to DNA damage response pathways, e.g., in a regulatory network.

Our results showed that the XRCC4 protein level strongly affects mRNA levels of the analysed BER and NHEJ genes. We suggested that the XRCC4 protein can possess the transcriptional regulation function either directly or indirectly. Indeed, searching the available data supporting this suggestion, we found “4 target genes of the XRCC4 transcription factor in ChIP-seq datasets from the ENCODE Transcription Factor Targets dataset” [84]. All these four genes are involved in RNA processing and nuclear structure organization and participate in cell signaling and differentiation. One of them, WDR74, is capable of modulating the RPL5–MDM2-p53 pathway and regulating the downstream transcriptional response, including p21 expression and cell cycle progression [85]. Therefore, XRCC4, as a probable transcriptional factor for WDR74, may affect the p53 level and cell cycle progression via the RPL5–MDM2–p53 pathway.

Querying six NHEJ and BER genes, *KU70*, *KU80*, *LIG4*, *XRCC1*, *LIG3*, and *APE1,* using the Enrichr/ChEA 2022 resource [86,87,88,89] revealed only one significant regulator (adjusted *p*-value = 0.0007909) common for all six genes: the MYC oncogene, a transcription factor with a wide array of functions affecting cellular activities such as the cell cycle, apoptosis, and DNA damage response [90]. We also carried the functional annotation of these NHEJ and BER genes using the STRING database of known and predicted protein–protein interactions [91]. This search provided certain publications supporting the associations between XRCC4 and MYC, although no direct experimentally verified interactions have been reported so far. The association between MYC and XRCC4 was discussed with relation to catastrophic genomic events in the condition of defective DNA damage repair [92]. Still, the specific pathways underlying such associations including multiple participants are yet to be identified. Figure 8 shows a summary of these database searches. For detailed explanations, the reader is referred to the cited references.

Additionally, we analysed the knockdown effect on the abundant nuclear enzyme PARP1 and found that it caused a twofold increase in the PARP1 protein in both normal and hTERT-modified cells. In earlier studies, PARP1 was mainly considered as a participant in the DNA repair process, specifically of the alternative NHEJ pathway. However, it has become clear that this enzyme is a multifunctional regulator of a variety of processes in the nucleus, including modulation of chromatin structure, transcription, replication, repair, and recombination [93,94]. PARP1 effects are mediated through binding its product PAR polymer to a variety of substrates, including histones. PARP1 can participate directly in the assembly of transcription complexes at enhancers and promoters. PARP1 participates in a signalosome complex containing DNA damage sensors ATM and PARP1 bound to IKKγ and PIASy in response to DNA damage [95]. It has been reported that, in response to low levels of DNA damage, PARP1 promotes cell survival and DNA repair, although heavy DNA lesions can result in parthanatos, a type of cell death mediated by an excess of PAR polymer [96]. In the context of the present study, it is important that PARP1 can also modulate Sp1 activity, affecting the cell cycle. It was described that increased PARylation suppressed Sp1 mediated transcription by preventing Sp1 binding to the Sp1 response element present in the promoters of target genes, such as *p21* and *p27*, resulting in inhibition of cell proliferation or G0/G1 cell cycle arrest [97].

Thus, an increase in PARP1 can produce a wide variety of effects. We suggest that here, this increase after XRCC4 knockdown is a part of the stress response, and we found it to occur independently of *hTERT* expression. This should be kept in mind when targeting XRCC4 in therapy and applying PARP1 inhibitors in anticancer therapy.

To evaluate the integral effect of XRCC4 knockdown on the cell transcriptional network, we analysed the expression of transcriptional factors p53 and Sp1. These two factors are often analysed in a pair because there is experimental evidence of their tight interaction in various regulatory networks and in gene transcriptional complexes [98]. On the other hand, their cooperation is rather complicated, because over 12,000 Sp1 and thousands of p53 binding sites are distributed throughout the human genome. Moreover, the possibility of their direct physical interaction with each other in promoters and multi-protein transcriptional complexes additionally complicates the question.

Here, we found that the protein levels of both transcription factors increased in response to XRCC4 knockdown in both normal TIG-1 and hTERT-modified NBE1 cells. In contrast, the effect on mRNA levels was not uniform between the cell lines and was not definite. The increase in the p53 protein could be predicted, since we disturbed the protein balance of the key enzymes involved in the DNA damage response, and p53 increases have been experimentally confirmed in other investigations on cellular stress and DNA damage response.

It is important that p53 can increase not only upon DNA damage, but also upon conditions that can lead to such damage, or other stress stimuli signaling harmful conditions for the cell. From this point of view, the observed p53 increase looks quite natural. It is known that the p53 protein level can be regulated in different ways. The primary regulatory pathway is the p53/MDM2 negative feedback loop. Within this pathway, p53 controls its own level via transcriptional activation of MDM2 expression [99,100]. In turn, MDM2 targets p53 protein for proteosomal degradation by poly-ubiquitination, thus keeping p53 protein levels relatively low. Upon cell stress, MDM2 and p53 are phosphorylated and bind to proteins that physically separate MDM2 from p53, resulting in an increased p53 protein level. Being activated, p53 either activates or represses a variety of target pathways involved in DNA repair, cell cycle arrest, apoptosis, autophagy, etc. The particular downstream pathway depends on many conditions, including the severity of the stress, the nature of the stressor, and the cell type.

However, the regulation of the p53 protein by the p53/MDM2 negative feedback loop does not explain the decrease in its mRNA levels observed in our experiments. Therefore, another pathway of p53 protein increase should be invoked. Within this pathway, the p53 protein level can increase as the result of unblocking its translation via release from inhibitory proteins that bind to 3′-UTRs [101]. In this case, the translation activation may lead to a decrease in mRNA level because of its degradation after utilization in translation. In general, once mRNAs enter the cytoplasm, they are either translated, stored for later translation, or degraded. Post-translation mRNA metabolism varies, but most of them degrade. This can explain why the p53 mRNA level did not remain unchanged after XRCC4 knockdown in our experiments, but decreased significantly. The obtained results demonstrate that p53 protein levels increased independently of p53 mRNA accumulation, thus indicating that the changes in the p53 protein level most probably resulted from protein stabilization.

The relative protein level of transcription factor Sp1 increased threefold after XRCC4 knockdown in both normal and hTERT-modified cell lines. It has been reported that Sp1 shares a common regulator with p53, MDM2, which targets Sp1 for proteasomal degradation [102]. Consequently, the protein levels of both transcription factors should change in the same direction, and we have observed this in our experiments.

On the other hand, Sp1 mRNA levels responded to XRCC4 protein elimination differently depending on the cell line. In normal TIG-1 cells, the mRNA level increased against the background of increased Sp1 protein and suppressed hTERT, while in hTERT-modified NBE1 cells, the mRNA level almost halved against the background of increased Sp1 and overexpressed hTERT. Clearly, hTERT interferes with the pathway of Sp1 transcriptional activation. 

According to the available knowledge, Sp1 transcription can be activated by many factors, including Sp1 protein itself. Additionally, the promoter activity can be affected by a number of proteins related to the cell cycle [103,104]. It was shown there that, in HeLa cells, overexpression of E2F, CDK4, cyclin D1, Rb and p21 increased Sp1 promoter activity and Sp1 mRNA levels, whereas increased p53 decreased Sp1 mRNA.

Since Sp1 is ubiquitously expressed and its activity can be modified by external stimuli and different stages of the cell cycle [105,106,107], it is rather beyond our limited experimental investigation to determine the specific participants in the pathway leading from XRCC4 protein decrease to Sp1 mRNA decrease. Nevertheless, our results indicate that hTERT can influence the Sp1 mRNA level and we believe these findings are essential for the adjustment of existing models of regulatory networks in normal and cancer cells.

It is known that at least three genes of the NHEJ system, DNA-PKcs, Ku70 and Ku86, have Sp1 binding sites in their promoters, and their protein levels decreased with the Sp1 decrease in human transformed kidney cell line 293T [108]. In the present study, we observed that the level of Ku70/Ku80 decreased after XRCC4 knockdown independently of the Sp1 increase. We suggest that this indicates that the final transcriptional effect is determined by higher levels of interactions between Sp1 and other transcriptional factors. 

As the transcriptional factor, p53 can either activate or repress its target genes. It is known that transcriptional activation occurs as the result of direct p53 binding to certain consensus sites in the regulatory regions of target genes, while the repression is an indirect process and occurs because of p53 interaction with activators or other mediators. The repression via interaction with other activator proteins has been reported for APE1, whose transcription is inhibited upon direct interaction of p53 with Sp1 bound to the APE1 promoter [109]. In our experiments, we observed the same negative correlation between the increase in p53 and the decrease in APE1 protein and mRNA levels.

The analysis of the cell cycle in TIG-1 cells demonstrated that normal cells responded to XRCC4 knockdown with a delay in G1/S phases, and this is consistent with the observed increase in p53 and its target protein, p21 [110,111,112,113,114]. On the contrary, in hTERT-modified NBE1 cells, XRCC4 knockdown led to a small but significant decrease in the G1 population, although the p53 protein was increased and there was slight but significant increase in cells in the S phase, thus apparently promoting G1/S transition. At the same time, the level of p21 protein responsible for G1 delay barely increased in hTERT-modified cells, indicating that other transcription factors may play essential roles in its activation [115].

The size of the G2/M population in hTERT-modified cells (6%) was nearly three times smaller than in normal cells (16%). XRCC4 knockdown did not significantly change the G2/M subpopulation in normal cells, although caused a slight increase in the G2/M subpopulation from 6% to 10% in NBE1 cells. We believe that the observed changes in cell cycle are the result of interplay between transcriptional factor (and other protein controlling mechanisms) interactions rather than the response to DNA damage. We found that normal cells responded to XRCC4 knockdown in the desired way, i.e., G1/S delay, while in the cells with hTERT overexpression, no such delay was induced.

In the present study, we also aimed to investigate how hTERT overexpression impacts the crosstalk between BER and NHEJ when targeting the NHEJ scaffold protein XRCC4. We found that, of all the analysed genes, the responses of two genes in hTERT-modified cells, i.e., Sp1 at the mRNA level and XRCC1 at the protein level, differed from normal cells after XRCC4 knockdown. As described above, XRCC4 knockdown caused a decrease in the Sp1 mRNA level in NBE1 cells in contrast to an increase in normal cells, although the Sp1 protein increased in both cell lines. It is known that the Sp1 protein is the activator for its own transcription; therefore, we believe that Sp1 transcription activation, which should have occurred under the increasing Sp1 protein, was overridden by the presence of overexpressed hTERT. The problem of crosstalk is further complicated by the fact that Sp1 and p53 together can modulate transcription of hTERT, whose promoter contains many sites for several transcription factors, including Sp1, while p53 is known to inhibit this activation [116,117,118,119].

The best-known function of human telomerase is maintaining telomeres, as overexpression of telomerase has been associated with cancer in almost 90% of cancers. Being the functional subunit of telomerase, hTERT catalyses the addition of nucleotides to the chromosome ends of telomeres, allowing the cells to become immortal. This fact is used to make immortalized cell lines, e.g., NBE1, which was used in this work. In normal cells, the activity of telomerase is suppressed, while in cancer, it is overexpressed. However, later, it became clear that maintaining telomeres does not fully explain the transformation of normal cells into cancer. Further investigations revealed an important role of hTERT in the transcriptional regulation of genes, in particular related to cell cycle progression, chromatin organization, and DNA damage response, although this function is less studied [120,121]. Thus, it has been reported that hTERT overexpression altered 284 genes associated with cell cycle regulation [122], metabolism, and apoptosis [123]. It has been reported that hTERT regulates VEGF, cyclin D1, and Mac-2BP [124,125]. The latter protein relates to the family of galectins implicated in modulating cell–cell and cell–matrix interactions. Yet another study demonstrated a link between hTERT and cell cycle progression through cyclin D1 protein levels that are dependent on the transcriptional regulation by hTERT [126]. Thus, we believe that the observed correlations are manifestations of the regulatory activity of hTERT and not solely the result of maintenance of telomeres by hTERT functioning in the telomerase complex.

## 4. Materials and Methods

### 4.1. Cells and Cell Culture 

The telomerase-immortalized cell line NBE1 with overexpressed hTERT (normal human bronchial epithelium cell line LIMM-NBE1 (RRID:CVCL_9Y83); https://www.cellosaurus.org/CVCL_9Y83 (accessed on 1 September 2024) was kindly provided by Prof. A. Ryan (University of Oxford). Normal human TIG-1 fibroblasts were obtained from the Coriell Institute Cell Repository (AG06173). Cells were cultured in Dulbecco’s modified Eagle’s medium with low glucose (DMEM, Life Technologies, Paisley, UK) supplemented with 15% foetal bovine serum (FBS, Life Technologies, Paisley, UK) at 37 °C in a humidified atmosphere with 5% CO_2_. Cells were routinely checked for mycoplasma. 

### 4.2. siRNA Transfections

siRNA transfections were carried out using the Lipofectamine RNAiMAX reagent (Life Technologies, #13778075) according to the manufacturer’s protocol. Unless otherwise indicated, cells were transfected with 33 nM siRNA and analysed 72 h after transfection. Control transfections were carried out using a non-targeting siRNA (Eurogentec, SR-CL000-005 Alnylam Pharmaceuticals, Cambridge, MA, USA). siRNA oligonucleotides (Table 1) were obtained from Eurogentec.

### 4.3. Real-Time PCR (qPCR)

Total RNA from cell pellets was isolated using the RNeasy Plus Mini Kit (Qiagen, Hilden, Germany) according to the manufacturer’s instructions. Sample quality and concentration were determined using a NanoDrop ND-1000 spectrophotometer (Thermo Scientific, Wilmington, DE, USA). Reverse transcription was performed using the “OT M-MulV-RH” kit (Biolabmix, Novosibirsk, Russia) on a Mastercycler Personal amplifier (Eppendorf, Enfield, CT, USA). Real-time PCR was performed using the “HS-qPCR SYBR blue (2×)” kit (Biolabmix, Novosibirsk, Russia). The comparative CT method was applied for the quantification of gene expression; *B2M* was used as the endogenous control. The sequences of primers used for qPCR are listed in Table 2.

### 4.4. Western Blot

Whole-cell extracts for Western blot analysis were prepared using a modified RIPA buffer. Cells collected with a spatula in PBS were centrifuged at 300× *g* for 3 min and resuspended in an equal volume of buffer I containing 10 mM Tris–HCl (pH 7.8), 200 mM KCl, 1 mg/mL of each protease inhibitor (pepstatin, aprotinin, chymostatin and leupeptin), 1 mM PMSF, 1 mM DTT, and 0.1 mM MG132. Then, a double volume of buffer II containing 10 mM Tris–HCl (pH 7.8), 600 mM KCl, 40% glycerol, 2 mM EDTA and 0.2% NP-40 (nonylphenoxypolyethoxyethanol, AppliChem, Ottoweg, Germany) was added, thoroughly mixed by pipetting, and incubated for one hour at 4 °C with constant stirring on a Bio RS-24 mini-rotator (BioSan, Riga, Latvia). The cell lysate was centrifuged at 13,600× *g* for 20 min at 4 °C in a 5424 R centrifuge (Eppendorf). The supernatant was collected into tubes and stored at −72 °C. The concentration of protein extracts was determined using the Bio-Rad Protein Assay kit (Bio-Rad, Hercules, CA, USA) according to the manufacturer’s protocol. Using an XCell SureLock Mini-Cell (Life Technologies, Wilmington, DE, USA), proteins were separated by vertical sodium dodecyl sulfate-polyacrylamide gel electrophoresis and transferred to a PVDF membrane (Merck Millipore, Carrigtwohill, Ireland), and the quality of the transfer was monitored by staining the membrane in Ponceau S (Medigen, Novosibirsk, Russia). Membrane blocking was performed in phosphate buffer solution (PBS, Medigen, PBS500) with 5% skim milk (AppliChem Panreac, Ottoweg, Germany). Antibody dilution and membrane washing were performed in PBS with 0.1% Tween-20 (BIO-RAD, #170-6531). Primary antibodies are listed in Table 3. Secondary antibodies conjugated with horseradish peroxidase were used for ECL reagent detection. Acquisition and densitometric quantification were carried out using an iBright™ FL1500 Imaging System. For quantification of relative protein levels, densitometric Western blot data were normalized to either actin or β-Tubulin.

### 4.5. Alkaline Comet Assay

Control and siRNA-transfected cells were trypsinized. The cell suspension (6 × 10^4^ cells/mL) was mixed with low-melting-point agarose (Bio-Rad, Hercules, CA, USA) at a final concentration of 0.75%. The suspensions were cast on microscope slides pre-coated with 1% regular agarose (Bio-Rad, Hercules, CA, USA) and allowed to set under cover slips on an ice-cooled metal plate. After solidification, the cover slips were removed and the slides were placed in the lysing solution (2.5 M NaCl, 100 mM Na_2_EDTA, 10 mM Tris, pH 10.5, 1% DMSO and 1% Triton X-100) overnight at 4 °C. Thereafter, the slides were placed in a horizontal gel electrophoresis unit filled with fresh electrophoretic buffer (1 mM Na_2_EDTA, 300 mM NaOH and 1% DMSO, pH > 13) and left in this buffer for 30 min at 4 °C in the dark. Without changing the alkali solution, the slides were electrophoresed for 30 min at 25 V (1.3 V/cm) at 4 °C in the Comet Assay Tank (Cleaver Scientific, Rugby, UK).

After electrophoresis, the slides were rinsed with 0.5 M Tris, pH 8.1 (3 × 5 min), and then stained with 1:10,000 SYBR S11494 Gold dye (Thermo Fisher Scientific, Wilmington, DE, USA) for 30 min (1 mL per slide).

### 4.6. Image Analysis

Pictures of 100 comets per slide were captured using an upright microscope platform Axio Imager 2, (Carl Zeiss MicroImaging GmbH, Jena, Germany). Image analysis of the data was performed using the Comet Analysis Software, version 1.0.0.0 (Trevigen, Minneapolis, MN, USA). The software automatically locates cells within the images and calculates tail length, percent DNA in the tail, and tail moment, among other parameters. The measure of damage was percent DNA in the tail (the integrated tail intensity times 100 divided by the total integrated cell intensity, providing a normalized measure of the percentage of total cellular DNA found in the tail). Data analysis was based on the mean population response. Plots were prepared with Excel.

### 4.7. Cell Cycle Analysis

Trypsinized cells were fixed in ice-cold 70% ethanol for at least 30 min. After removal of the fixation solution by centrifugation, cells were washed with PBS, treated with 100 µg/mL of DNase free RNase A (Thermo Scientific, EN0531), 0.1% Triton ×100 (AppliChem Panreac, A4975) at 37 °C in PBS for 30 min, and further stained with 10 µg/mL propidium iodide (Sigma, St. Louis, MO, USA). Samples were run on a BD FACSAria III (BD Life Sciences, San Jose, CA, USA) and the cell cycle distribution was analysed using BD FACSDiva Software v9.0 (BD Biosciences).

### 4.8. Statistical Analysis

Statistical analysis was performed using Student’s *t*-test and Microsoft Excel 2013. Sample size is indicated for each experiment.

## 5. Conclusions

There is bidirectional crosstalk between the BER and NHEJ systems. The deficiency in NHEJ induced by knockdown of its scaffold protein, XRCC4, compromised BER in both normal cells and cells overexpressing hTERT, although cells with overexpressed hTERT retained their BER scaffold protein XRCC1. Analysis of the impact of XRCC4 knockdown on mRNA levels of the studied genes of BER and NHEJ highlighted the functions of this scaffold beyond a mere docking platform for tethering components in a multiprotein complex, and this implies its connection to a transcriptional regulatory network or mRNA metabolism. XRCC4 knockdown increased ubiquitous transcription factors Sp1 and p53, but caused cell G1/S delay only in normal cells, with the opposite effect in cells with overexpressed hTERT. The knockdown of one scaffold of one specific DNA repair system triggered a far-reaching response beyond the DNA repair systems, highlighting that the crosstalk between BER and NHEJ systems is an integral part of the regulatory network of the cell.

## Data Availability

The original contributions presented in the study are included in the article/Appendix A, further inquiries can be directed to the corresponding author/s.

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
