# Peer review of "Crosstalk between BER and NHEJ in XRCC4-Deficient Cells Depending on hTERT Overexpression"

_ijms, 2024, doi:10.3390/ijms251910405_

Round 1

Reviewer 1 Report

Comments and Suggestions for Authors

Thank you for submitting your manuscript for consideration at IJMS

the idea is novel and you described it nicely however here are my comments

1-Having a graphical abstract or figure that explains and summarizes your findings is highly recommended.

2- Figure 8 shows only histograms without actual western blots

3- Figure 9 although showed a significant increase in P53 protein levels the increase is subtle and does not match with mRNA levels and this needs more explanation in the discussion part

Comments on the Quality of English Language

Minor typos were found

Reviewer 2 Report

Comments and Suggestions for Authors

Comments to author:

The manuscript of an article, which was written by Dr. Svetlana V. Sergeeva et al, is interesting, discussing alterations in BER- and NHEJ-associated gene expression by introduction of XRCC4 si RNA in hTERT normal and overexpressing cells. However, I would suggest author to edit text and Figures. The most important is to edit text avoiding redundant expression and combine Figures to make the manuscript to be as concise as possible but remained with essential discussion.

Recommendation: Major revision

General comments

  The main concern about presentation of the data is that summarizing results from the same experimental settings. Reviewing the manuscript, I understood that Figures 1 to 9, and 11 were obtained from the same RNA-/protein-containing samples from TIG-I or LIMM-NBE1 cells. At least, Figures 1-3 and Figures 4-6 could be combined. Moreover, I found that the histogram of the siCtrl, indicating 1.00 without an error bar, in each Figure could be eliminated but commented in each Figure legend. I recommend the normalization could be indicated as dotted lines. That will greatly reduce the space for Figures, showing essential results.

  The secondary important is that the reasons why gene expression of KU70, KU80, LIG4, XRCC1, LIG3, and APE1 in TIG-1 and LIMM-NBE1 cells were reduced should be discussed. Aren’t the results just indicating that were due to the off-target effect with siXRCC4? Regarding to this problem, authors had better to confirm 5’- or 3’-UTRs of the genes that are regulated in accordance with the knock down of the expression of the XRCC4 gene.

  The third is, the blots of proteins, KU70, KU80, LIG4, XRCC1, LIG3, APE1, p53, Sp1, PARP1, and p21 are not indicated in Figures 2, 3, 5, 6, 8, 9, and 11. Are they surely indicating each protein level? If they were quantified by another methods, including ELISA, it should be noted.

  Finally, it is obscure how hTERT affected expression of the BER- and NHEJ-associated genes during siXRCC4 treatment. Comparing Figures 2 with 5 or Figures 3 with 6, I do not find remarkable differences between them.

Specific comments

P1, Abstract and all through the text: In NBE1 or LIMM-NBE1 cells, how hTERT was modified? From the text, I thought it overexpress hTERT. How about the telomerase activity in NBE1 or LIMM-NBE1 cells? Firstly, authors need to describe the differences between these two cell lines.

P3, Figure 1 and other Figures: The vertical axis, Transcript/Protein Fold changes need to be defined. Authors had better describe how they were normalized (standardized) and how they were compared with the values from siRNA-introduced cells.

P3, L102: constitute a; Why not in the plain text?

P3, L105 and L118: 72 hours/72hrs; unify all through the text.

P3, L111: no results with *p<0.05?

P3, Figure 3 light panel: I recommend authors add Mw.

P3, L114: …Ku70/ Ku80 and ligating enzyme Lig4. Lig4 functions…; …Ku70/Ku80 and Lig4, which functions…(I recommend authors check all through the similar expression all through the text.)

P8, Figure 7: It needs to be redrawn. Give (C) and (D) for TIG-1 siXRCC4, and NBE1 siXRCC4, respectively. Details should be described on the legend. Then, (E) would be the quantification of the results. It needs more about the description for the evaluation. %DNA in Tail should be placed on the vertical axis. What it means with the minus value, additionally?

P8, L215: XRCC4 knockdown affect on…; XRCC4 knockdown induces…

P8, L216: …PARP1 was believed the…; PARP1 was believed to be the…

P9, 10, and 12, Figures 8, 9, and 11: These Figures can be combined with. Eliminate histograms for siCtrl, providing the definition of the fold changes on the legend.

P13, L376: ?-XRCC1; delete ?

P13, L379: delete the extra space. Check other parts.

P17, 4.1. Cell culture: should be “Cells and Cell culture”, providing details about TIG-1 and LIMM-NBE1 cells. If NBE1 is the abbreviation of LIMM-NBE1, unify all through the text.

P17 Materials and Methods: Although a sequence of the miXRCC4 is indicated, that of the siXRCC and the siCtrl are not.

P17, L608: Primers used for what?

P12 to 17, Discussion: is too long. It should be summarized within three pages, eliminating redundant expression but remain essential descriptions. The main thing is to discuss on the effect of siXRCC4 and hTERT on the expression, including BRE-/NHEJ-associated genes.

Comments on the Quality of English Language

See Comments and Suggestions for Authors

Round 2

Reviewer 2 Report

Comments and Suggestions for Authors

Comments to authors:

The manuscript, which was written by Dr. Svetlana V. Sergeeva et al has been almost successfully revised according to the suggestions that I have commented. I also found that the title has been edited. Importantly, authors have edited the text to easily understandable Figures with minimal but essential information. In addition, I evaluate that authors have successfully summarized the text as concise as it could be. In the revised version, I just encourage author to edit the text adding essential descriptions on the reasons why gene expression of KU70, KU80, LIG4, XRCC1, LIG3, and APE1 in TIG-1 and LIMM-NBE1 cells were reduced. Authors answered that it was not likely due to the off-target effects of the introduction of the siRNAs, neither the effects on 3’-UTRs. Was it examined if there are common transcription factor-binding motifs in their 5’-upstreams in their promoter regions? I recommend authors to make a list that summarizes possible reasons with just essential references. Additionally, authors had better check all through the text again to confirm that it is completely freed from any errors, including typos and formatting of the references.

Recommendation: Accept after minor revision

Round 3

Reviewer 2 Report

Comments and Suggestions for Authors

Comments to authors:

The manuscript, which was written by Dr. Svetlana V. Sergeeva et al has been much improved comparing with the original one. The effort to discuss XRCC-mediated molecular mechanisms to regulate BER/NHEJ-associated proteins in TERT-overexpressing cells should be evaluated. The right discussion might lead authors to find and show the biologically meaningful signal pathways to evoke DSB responses. Next step will be that authors prove the concept of Figure 8 experimentally.

Recommendation: Can be accept as it is